# Impact of Impulsivity and Therapy Response in Eating Disorders from a Neurophysiological, Personality and Cognitive Perspective

**DOI:** 10.3390/nu14235011

**Published:** 2022-11-25

**Authors:** Giulia Testa, Roser Granero, Alejandra Misiolek, Cristina Vintró-Alcaraz, Núria Mallorqui-Bagué, Maria Lozano-Madrid, Misericordia Veciana De Las Heras, Isabel Sánchez, Susana Jiménez-Murcia, Fernando Fernández-Aranda

**Affiliations:** 1Universidad Internacional de La Rioja, 26006 La Rioja, Spain; 2CIBER Physiology of Obesity and Nutrition (CIBEROBN), Carlos III Health Institute, 28029 Madrid, Spain; 3Department of Psychobiology and Methodology, Autonomous University of Barcelona, 08193 Barcelona, Spain; 4Psychoneurobiology of Eating and Addictive Behaviors Group, Institut d’Investigació Biomèdica de Bellvitge (IDIBELL), 08907 Barcelona, Spain; 5Department of Psychiatry, University Hospital of Bellvitge, 08907 Barcelona, Spain; 6Addictive Behaviours Unit, Department of Psychiatry, Hospital de la Santa Creu i Sant Pau, Biomedical Research Institute Sant Pau (IIB Sant Pau), 08041 Barcelona, Spain; 7Department of Psychology, University of Girona, 17004 Girona, Spain; 8Neurophysiology Unit, Neurology Department, Hospital Universitari de Bellvitge, 08908 L’Hospitalet de Llobregat, Spain; 9Department of Clinical Sciences, School of Medicine and Health Sciences, University of Barcelona, 08907 L’Hospitalet de Llobregat, Spain

**Keywords:** eating disorders, impulsivity traits, inhibitory control, event-related potentials, treatment outcome

## Abstract

Impulsivity, as a multidimensional construct, has been linked to eating disorders (EDs) and may negatively impact treatment response. The study aimed to identify the dimensions of impulsivity predicting poor remission of ED symptoms. A total of 37 ED patients underwent a baseline assessment of impulsive personality traits and inhibitory control, including the Stroop task and the emotional go/no-go task with event-related potentials (ERPs) analysis. The remission of EDs symptomatology was evaluated after 3 months of cognitive-behavioral therapy (CBT) and at a 2-year follow-up. Poor remission after CBT was predicted by poor inhibitory control, as measured by the Stroop task. At 2 years, the risk of poor remission was higher in patients with higher novelty seeking, lower inhibitory control in the Stroop and in ERPs indices (N2 amplitudes) during the emotional go/no-go task. The present results highlight inhibitory control negatively impacting both short- and long-term symptomatology remission in ED patients. On the other hand, high novelty seeking and ERPs indices of poor inhibition seem to be more specifically related to long-term remission. Therefore, a comprehensive assessment of the impulsivity dimension in patients with ED is recommended to tailor treatments and improve their efficacy.

## 1. Introduction

Impulsivity is recognized as a multidimensional construct, reflecting multiple and separable psychological dimensions. An important contribution to this multidimensional conception has been provided by the UPPS model, which describes different personality traits that reflect impulsive behaviors [1].

From a neuropsychological perspective, the cognitive functions linked to impulsivity include inhibitory control and decision-making processes [2]. Inhibitory control refers to the ability to inhibit cognitive or motor responses [3,4,5]. Cognitive inhibition is usually measured by interference control tasks (e.g., Stroop Color–Word Task) in which effortful inhibition at a covert cognitive level is required to suppress the competing automatic response in favor of the correct response [6]. In contrast, inhibition of motor responses is assessed by go/no-go tasks, which measure the overt effortful expression of inhibition, involving the suppression of activated motor response [6].

Impulsivity has been proposed as a transdiagnostic feature of eating disorders (EDs) [7,8]. From a traditional view, impulsivity mainly characterizes the bulimic EDs spectrum, including binge eating disorder (BED) and bulimia nervosa (BN), whereas compulsivity would be more likely to be associated with anorexia nervosa (AN) [9,10]. However, evidence of impulsivity also exists in patients with AN [11,12], which is in line with the transdiagnostic approach of EDs [13,14].

Impulsivity plays a role in the etiology and maintenance of ED symptoms, which may have important implications for therapy response. Cognitive-behavioral therapy (CBT) is one of the most common and effective treatments, which have been shown to reduce EDs symptoms [15]. However, a considerable number of patients are at risk of dropping out of therapy, and others do not show complete remission of symptoms after CBT [16]. Identifying the factors that interfere with the optimal remission of EDs symptoms following CBT and at a longer follow-up is crucial to designing more personalized treatment approaches [17].

Although impulsivity is not necessarily dysfunctional in the nonclinical population [18], in individuals with mental disorders, impulsive personality traits predicted poor treatment outcomes [19,20]. Similarly, in individuals with EDs, impulsivity has been related to lower engagement with treatment and higher dropout rates [21,22,23]. Novelty seeking has been associated with a higher risk of dropout and not obtaining full remission [24]. Negative urgency, which reflects the tendency to rush impulsively in response to negative emotions, has also been shown to predict poor treatment outcomes in patients with BED [25].

Regarding inhibitory control, studies in patients with EDs have shown difficulties in both motor and cognitive inhibition [26,27,28,29], even though some discrepancies are present in the literature [30,31]. Furthermore, poor inhibition has been suggested to interfere with treatment remission in individuals with substance addictions [32] and behavioral addictions [33]. Similarly, low inhibitory control predicted poor weight loss after treatment in individuals with obesity [34,35,36].

Previous studies showed the association between poor decision making and treatment outcome in EDs [37,38]. However, the impact that inhibitory control may have on therapy response in EDs is heterogeneous and may need further research [25,39]. The recording of electroencephalographic (EEG) activity during response inhibition tasks gives more sensitive indices of inhibitory control through the analysis of event-related potentials (ERPs). Specifically, the ERP component classically associated with inhibition is the N2, which is a negative wave that emerges approximately 200–300 ms after stimulus presentation. The amplitude of the N2, which is usually enhanced in “no-go” compared to “go” stimuli, is a valuable measure of inhibitory control [40]. Lack of inhibitory control indexed by the N2 amplitude has been reported among clinical samples, including individuals with substance addiction [41,42], behavioral addiction [43] and BED [44]. However, further research is needed to determine whether a lack of inhibitory control is related to treatment response in individuals with EDs.

The present study aimed to analyze the impact of several facets of impulsivity in ED therapy response. Within this scope, multiple components of impulsivity were evaluated in a comprehensive perspective, including: (1) impulsive personality, measured by the UPPS dimensions and novelty seeking trait; (2) inhibitory control process, measuring cognitive inhibition with a Stroop task; and (3) motor inhibition, measured with an emotional go/no-go task with EEG recording.

This multidimensional impulsivity assessment was conducted at baseline, and regression models were adopted to identify which dimensions predicted ED symptomatologic remission after CBT treatment and at a longer follow-up (i.e., 2 years). We hypothesized that the most impulsive individuals would present partial or non-remission of EDs symptomatology, after CBT and at the longest follow-up. Different dimensions of impulsivity are expected to contribute to suboptimal remission of EDs symptoms, observed both immediately after treatment and in the medium term.

## 2. Materials and Methods

### 2.1. Participants

A total of 37 treatment-seeking individuals with ED were consecutively recruited at the ED Unit within the Department of Psychiatry at Bellvitge University Hospital (HUB)—a public health hospital certified as a tertiary care center with a highly specialized unit for the treatment of ED in Barcelona (Spain). To avoid the possible gender difference in impulsivity shown in the literature [45], recruitment was limited to female patients, which is the most representative gender in EDs. Patients were diagnosed with anorexia nervosa (AN; *n* = 20) and bulimic spectrum disorders (BSD *n* = 17), including bulimia nervosa (BN) and binge eating disorder (BED), according to DSM-5 criteria [46]. Patients voluntarily participated in the study, and their informed consent was obtained. They all underwent a baseline assessment before starting the CBT treatment. Remission of ED symptoms was analyzed after CBT and in a two-year follow-up. To that end, clinical records and online shared electronic medical records were analyzed retrospectively throughout Catalonia (Spain).

### 2.2. Measures

#### 2.2.1. Baseline Assessment

The Temperament and Character Inventory-Revised (TCI-R) [47] validated for the Spanish population [48] is a questionnaire of 240 items answered on a 5-point Likert scale. The novelty seeking subscale of the TCI-R was adopted in the present study as a measure of impulsive temperament. The internal consistency of the subscale in the sample was α = 0.836.

The UPPS-P Impulsivity Scale [49] is a 59-item scale that assesses impulsive behavior on 5 different scales: sensation seeking, lack of premeditation, lack of perseverance, negative and positive urgency. Positive urgency has been included more recently. All items are rated on a 4-point scale from 1 (strongly agree) to 4 (strongly disagree). The UPPS-P has satisfactory psychometric properties in terms of both convergent and discriminative validity. The Spanish version of the UPPS-P scale was obtained by a back-translation process, and its Spanish adaptation shows adequate psychometric properties [50]. The α values for the different UPPS-P scales in our sample are lack of premeditation (0.836), lack of perseverance (0.850), sensation seeking (0.827), positive urgency (0.941) and negative urgency (0.861). Total score (0.911).

The Eating Disorders Inventory (EDI-2) [51] was adopted to screen symptomatology related to eating disorders on a six-point Likert scale. EDI-2 is a self-report measure consisting of 91 items and provides scores on 11 subscales: drive for thinness, body dissatisfaction, bulimia, ineffectiveness, perfectionism, interpersonal distrust, interoceptive awareness, maturity fears, asceticism, impulse regulation and social insecurity. The sum of all subscales provides an eating disorder measure, which is considered a global scale of ED severity. The internal consistency of the global scale in the sample was α = 0.931.

The Stroop Color–Word Test (SCWT) [52] is a paper and pencil test, which measures the ability to inhibit cognitive interference known as the Stroop effect. The task consists of reading 3 pages with 100 words each as fast as possible. The first 2 pages are called the “congruous condition”, and the participants are asked to (1) read the color words printed in black and (2) name the colors of the printed “Xs” (red, green and blue). The last page (3) contains the names of colors printed in an incongruent color (i.e., the word ‘‘red’’ printed in blue ink), and the subjects are asked to name the color of the ink instead of reading the word. The subjects are given 45 s for each page, and when the time is over, the last named item is noted. The total score for each task is calculated from the number of items completed on each page. Higher scores in the incongruent task variable indicate a better capacity for inhibition response. The test has shown adequate reliability and construct validity for the assessment of inhibition and switching skills [52].

The emotional go/no-go task [8] is a computerized task for assessing response inhibition. Participants were presented with 600 images surrounded by a colored frame. They were asked to respond as quickly as possible to images within a blue frame (i.e., go cues) and to withhold the response to images within a yellow frame (i.e., no-go cues). The images were divided into three blocks presented in a randomized way and with different emotional valence: 200 pleasant images, 200 neutral images and 200 unpleasant images. Out of each block of 200 images, 75% were go cues, and 25% were no-go cues. The interstimulus interval was pseudorandomized from 1.500 to 1.700 ms to discourage anticipatory responses. The reaction times (RTs) in go trials and the accuracy in go and no-go trials were calculated for each emotional category.

The electroencephalogram (EEG) was recorded continuously throughout the emotional go/no-go task using PyCorder (BrainVision). In total, 60 active Ag/AgCI electrodes were placed into an EEG recording cap (EASYCAP GmbH), distributed according to the 10–20 system; additional 3 electrodes were adopted for recording the vertical and horizontal electrooculogram (EOG), and Cz was used as an online reference. Impedances were kept below 20 kΩ using the SuperVisc high-viscosity electrolyte gel for active electrodes. Signals from all channels were digitized with a sampling rate of 500 Hz and 24 bit/channel resolution and online filtered between 0.1 and 100 Hz. Offline EEG analyses were performed with Brain Vision Analyser consisting of the following steps: high pass filtering at 0.1 Hz, low pass filtering at 30 Hz (Butterworth zero-phase filter; 24 dB/octave slope) and a notch filter at 50 Hz; raw data inspection for manual detection of artifacts and screening for bad channels, semi-automatic eye-blink correction using independent component analysis (ICA); artifact rejection of trials with an amplitude exceeding ±80 μV; EEG data were segmented into 1500 ms epochs from 500 ms before to 1000 ms after stimulus onset. Data were baseline corrected against the mean voltage during the 200 ms pre-stimulus periods. Artifact-free epochs were separately averaged for each subject in each experimental condition (go, no-go) and stimulus type (positive, negative, neutral). Event-related potentials (ERPs) analyses were based on visual inspection of the grand average waveforms and the existing literature. Peak amplitudes for the N2 were analyzed in a frontocentral electrodes cluster (FC1, FC2, Fz, C3, C4, Cz), in time windows between 200 and 380 ms. Since N2 is a negative peak wave, the more negative the values, the greater its amplitude.

#### 2.2.2. Treatment

Patients received cognitive-behavioral therapy (CBT) at HUB, which was carried out by clinical psychology experts in the field. Patients diagnosed with AN completed a day hospital treatment program, which included group CBT sessions, lasting 90 min each, for 15 weeks. Treatment for the other EDs diagnoses (BED, BN) consisted of 16 weekly outpatient group sessions of CBT lasting 90 min. All patients attended follow-up sessions for a period of about two years’ duration. The goal of the treatment was to train patients to implement CBT strategies to reduce eating symptoms and to enable patients to acquire good healthy habits. Voluntary treatment discontinuation was categorized as dropout (i.e., not attending treatment for at least three consecutive sessions). Patients completing treatment were re-evaluated by their clinician to classify the remission of ED-related symptomatology. According to the DSM-5 criteria [46], full remission was considered as the total absence of ED symptoms meeting diagnostic criteria for at least 4 consecutive weeks. We considered full remission as an index of good treatment outcomes. By contrast, we considered the following as measures of “poor treatment” outcomes: voluntary treatment discontinuation or dropout (i.e., not attending treatment for at least three consecutive sessions); partial remission of EDs (i.e., symptomatic improvement with residual symptoms); and non-remission of EDs.

### 2.3. Statistical Analysis

Statistical analysis was carried out with Stata17 for Windows [53]. The comparison between the groups defined for the treatment outcome (good versus poor) was performed with an analysis of covariance (ANCOVA) adjusted for the ED subtype. Finner’s method controlled the increase in type I error due to the multiple null-hypothesis tests [54], and the effect size of the mean differences was estimated with the standardized Cohen d-coefficient (mild–moderate effect size was considered for |d| > 0.5 and large–high effect size for |d| > 0.8) [55].

Two predictive models were obtained for the risk of poor treatment outcome (defined as the dependent variable, with values 1 = good outcome versus 0 = poor outcome) post-CBT and at 2-year follow-up, with logistic regression models adjusted for the ED subtype. The list of the potential predictors included the EEG measures registered during the emotional go/no-go task, the Stroop interference score and the impulsivity scores (obtained in the UPPS-P and the NS scales). Goodness of fit was assessed with the Hosmer–Lemeshow test (*p* > *0*.05 is indicative of adequate fit) [56], predictive capacity with the Nagelkerke pseudo-R2 coefficient and global discriminative capacity with the area under the ROC curve (AUC).

## 3. Results

### 3.1. Characteristics of the Participants

The first block of Table 1 displays the description of the sociodemographic variables. Most participants were single (62.2%), with secondary education (51.4%), employed (54.1%) and within mean-low to low social position indices (75.7%). The mean age was 30.7 years (SD = 12.0); the mean age of onset of ED-related problems was 22.2 years (SD = 8.4); and the mean duration of the disease was 8.5 years (SD = 8.4).

### 3.2. Variables Related to the CBT Outcome and Follow-Up

The second block of Table 1 displays the distribution of treatment outcomes during the CBT treatment and at 2-year follow-up. Good remission was achieved for 40.6% of the participants at the end of the treatment plan (the risk of poor outcome was 59.4%). At the 2-year follow-up, good remission was registered for 43.3% of the sample (the risk of poor outcome was 56.7%).

Table 2 shows the results of the ANCOVA exploring the relationships between the clinical variables measured at baseline (duration of the ED, EDI-2 scales, UPPS-P scales, TCI-R novelty seeking, go/no-go task and Stroop interference) and the treatment outcome (good/poor) measured at two time points—(1) at final treatment after CBT and (2) at 2-year follow-up. These analyses were adjusted–controlled for the ED subtype. At the end of the CBT (i.e., final treatment), patients with poor outcomes were characterized by higher scores in the EDI-2 bulimia and interpersonal distrust, higher values in UPPS-P sensation seeking, lower values in the accuracy go task and a lower average score in Stroop interference (these measures registered a difference in the significance test and/or effect size within the ranges mild to large).

At the 2-year follow-up, poor outcome was related to higher mean scores in the EDI-2 bulimia and impulse regulation scales, higher personality traits related to impulsivity (in the UPPS-P total and TCI-R novelty seeking scales), lower mean in the Stroop interference score and lower amplitude of the N2 wave in positive go trials, negative no-go trials and neutral go trials (mean differences with a significant result and/or effect size within at least the mild range). The ERPs and the topographical maps for the no-go negative condition are presented in Figure 1, showing the lower N2 amplitude in patients with poor outcome at follow-up compared to those with good outcome. Figure 2 displays the radar charts with the results of the comparisons between the patients with good and poor treatment outcome (z-standardized means are plotted due to the different measurement scales of the variables analyzed in the study).

The two logistic regression models displayed in Table 3 indicated that a lower score in the Stroop interference task increased the risk of poor outcomes post-CBT. At the 2-year follow-up, the risk of poor outcomes was increased for patients with higher scores in the TCI-R novelty seeking scale, lower amplitude of the N2 in negative no-go trials and lower scores in the Stroop interference.

## 4. Discussion

The present study investigated the impact that impulsivity may have on therapy response in patients with EDs. Remission of ED symptomatology was evaluated after 3 months of CBT and at a 2-year follow-up. Multiple impulsivity dimensions were measured before treatment and significantly predicted poor treatment response, with more consistent evidence emerging at follow-up.

With regard to impulsive personality traits, patients with poor remission following CBT were characterized by higher sensation seeking, whereas those with poor remission at follow-up were characterized by higher novelty seeking and higher UPPS total score. Novelty seeking emerged as a predictor of poor ED remission at follow-up, suggesting its relation to long-term remission of ED symptomatology. Along this line, some of the previous studies in patients with EDs showed associations between novelty seeking and a higher risk of dropout and suboptimal remission following CBT [24,57].

At a neurocognitive level, lower cognitive control predicted CBT outcome and remission at follow-up. Thus, individuals with difficulties in controlling the interference in the Stroop task showed poor remission of EDs after 3 months of CBT and at 2 years from treatment. In contrast, a study in patients with BED did not show an association between cognitive inhibition and reduction in ED psychopathology after treatment [39]. The discrepant results are possibly related to the heterogeneity of the present sample, which includes various ED subtypes other than BED. Interestingly, reduced cognitive inhibition has been associated with longer EDs duration in AN [58], which in turn is a factor contributing to poor remission of EDs. The present results in a mixed sample of EDs suggest that cognitive inhibition is a relevant factor associated with both short-term and long-term remission. Future studies in larger samples would help detect and elucidate the differences across ED subtypes.

Regarding motor inhibition, the lower amplitude of the N2 was associated with poor ED remission at follow-up. By contrast, the behavioral measures of response accuracy in the go/no-go task did not predict EDs remission, in line with some previous findings [25]. It can be argued that the ERPs indexed, such as the N2, may be particularly sensitive in detecting alterations in inhibitory processes, as recently shown in patients with BN undergoing an odd-ball task [59]. So far, this is the first evidence of a relation between lower N2 amplitude and suboptimal remission of ED symptoms. Interestingly, this effect was shown to be maximal in the no-go trials with negative emotional stimuli. Affective versions of response inhibition tasks (e.g., emotional go/no-go) have been adopted to study how response inhibition is modulated by emotional stimuli [60]. In this case, the difficulties in inhibition (indexed by lower N2 amplitudes) when a negative emotional state is induced could be more strongly related to poor remission of EDs. This result may be explained by the fact that both impulsivity and emotion regulation difficulties have been proposed as central transdiagnostic phenomena across EDs [61,62].

Taken together, the present results highlighted the relevance of novelty seeking and inhibitory control in remission of EDs. A comprehensive assessment of impulsivity, including personality traits and neurocognitive indices of inhibitory control, may be particularly useful to improve treatment effectiveness. For instance, those individuals with a tendency to be more impulsive, excitable, dramatic and with intolerance to routine might benefit from treatment tailored to reduce their impulsive behaviors. To address the difficulties in inhibition, inhibitory control training with general or food-specific stimuli has been tested in individuals with EDs showing promising results [63,64]. Recently, the outcome of a food-specific inhibitory control training has been measured with ERPs indices [65]. The results of our emotional go/no-go task encourage testing the effectiveness of novel inhibitory training using emotional stimuli to target impulsivity and emotion regulation.

The present findings should be considered under some limitations. First, the small sample was not suitable for the analysis of different ED subtypes, even though this variable was controlled for by statistical adjustment. Nevertheless, it could be of interest to study in the future the relationship between impulsivity dimensions and remission in specific ED subgroups. The small sample size also impacts the ability to avoid type II errors, the power capacity to detect the existence of a true relationship and the accuracy of the results obtained in the multivariate analyses. In this sense, the empirical evidence of this work should be interpreted with caution, pending future studies with larger samples to confirm or refute it. In addition, it should be considered that the assessment conducted in this study is difficult to perform in clinical samples, and therefore, research in this area is scarce and with low sample sizes. A second limitation is related to the nature of the outcome measure. Specifically, ED remission was assessed by the clinician at the end of treatment and at follow-up, according to the DSM-5 criteria. It is important to remark that optimal remission included only those individuals who fully remitted from symptomatology. Although this is particularly relevant in clinical practice, the adoption of a quantitative measure to track the changes in ED symptoms (e.g., EDI-2) should be considered in future works. Finally, the absence of a control group of patients undergoing treatment other than CBT or untreated individuals limits the interpretation and generalization of these results.

Despite these limitations, several strengths of the study should be remarked on, such as the comprehensive assessment of impulsivity, which included personality, neuropsychological and neurophysiological measures. Thus, the multidimensional assessment enables a better characterization of impulsive profiles, which could interfere with treatment outcomes. Furthermore, remission of ED was not only considered immediately after treatment but also at a longer follow-up of 2 years, thus providing an opportunity to study the relationship between impulsivity and long-term recovery.

## 5. Conclusions

In conclusion, high novelty seeking and low inhibitory control in individuals with EDs, but also specific neurophysiological indices, seem to contribute to poor remission of ED symptomatology. In particular, cognitive inhibition emerged as the dimension of impulsivity that more consistently predicted both short-term and medium-term remission of ED symptoms, confirming the importance of conducting a comprehensive assessment.

From a clinical perspective, early detection of patients with a lack of inhibition is recommended to personalize treatments and improve their effectiveness. A replication of these results in individuals with different subtypes of EDs is needed in future studies assessing the response to treatment.

## Figures and Tables

**Figure 1 nutrients-14-05011-f001:**
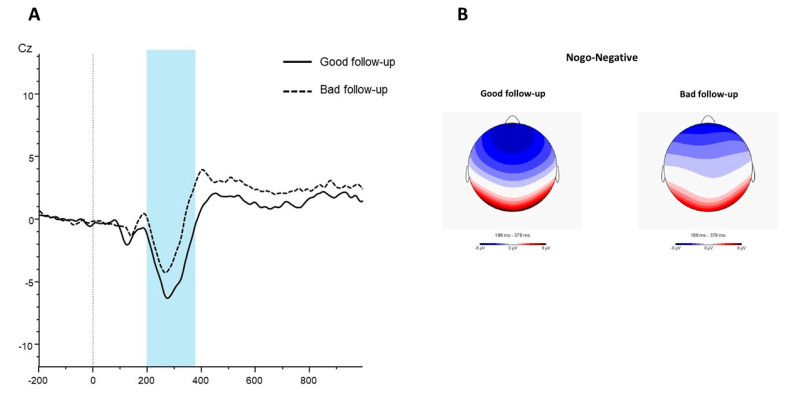
(**A**) Grand average ERPs waver for the no-go negative condition in the group of patients with good outcomes at follow-up (continuous line) and those showing bad outcomes at follow-up (dotted line). (**B**) Topographical maps (200–380 ms) for the no-go negative condition in patients with good outcomes at follow-up (**left panel**) and those with poor outcomes at follow-up (**right panel**).

**Figure 2 nutrients-14-05011-f002:**
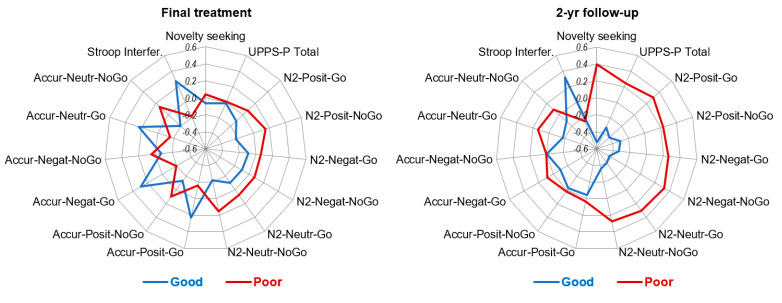
Radar charts with the z-standardized mean scores obtained among the groups with good and poor treatment outcomes.

**Table 1 nutrients-14-05011-t001:** Description of the sample.

Sociodemographic	*n*	%	Clinical Profile	Mean	SD
Civil status			Age (years old)	30.73	12.00
Single	23	62.2%	Age of onset of ED (years old)	22.22	8.42
Married	12	32.4%	Duration of ED (years)	8.48	8.41
Divorced	2	5.4%	ED subtype	*n*	*%*
Education			Anorexia nervosa	20	54.1%
Primary	11	29.7%	Bulimia nervosa	17	45.9%
Secondary	19	51.4%	Treatment outcome: end treatment	*n*	*%*
University	7	18.9%	Dropout	8	21.6%
Employment			Non-remission	1	2.7%
Employed/student	20	54.1%	Partial remission	13	35.1%
Unemployed	17	45.9%	Full remission	15	40.5%
Social position			Treatment outcome: 2-year follow-up		
High	1	2.7%	Dropout	9	24.3%
Mean-high	5	13.5%	Non-remission	3	8.1%
Mean	3	8.1%	Partial remission	9	24.3%
Mean-low	9	24.3%	Full remission	16	43.2%
Low	19	51.4%			

Note. SD: standard deviation.

**Table 2 nutrients-14-05011-t002:** Variables related to the treatment outcome: ANCOVA adjusted for ED subtype.

	Final Treatment						2-Year Follow-Up					
	Good (*n* = 15)		Poor (*n* = 22)				Good (*n* = 16)		Poor (*n* = 21)			
	Mean	SD	Mean	SD	*p*	|d|	Mean	SD	Mean	SD	*p*	|d|
Duration of ED (years)	8.62	10.76	8.39	6.46	0.928	0.03	8.60	10.68	8.39	6.43	0.933	0.02
EDI-2 Drive for thinness	11.22	5.64	12.08	6.15	0.669	0.15	12.13	5.85	11.43	5.99	0.718	0.12
EDI-2 Body dissatisfaction	16.76	7.48	16.08	6.80	0.742	0.10	16.36	6.76	16.34	7.50	0.993	0.00
EDI-2 Interoceptive awareness	12.09	7.12	10.58	5.56	0.489	0.24	12.87	7.43	9.91	4.83	0.155	0.47
EDI-2 Bulimia	4.41	4.68	7.77	5.40	0.025 *	0.67 ^†^	4.65	4.44	7.74	5.43	0.034 *	0.62 ^†^
EDI-2 Interpersonal distrust	3.97	3.50	6.97	5.48	0.080	0.65 ^†^	5.98	4.93	5.58	5.02	0.814	0.08
EDI-2 Ineffectiveness	11.95	6.98	9.81	5.69	0.333	0.34	10.99	6.36	10.44	6.30	0.799	0.09
EDI-2 Maturity fears	8.98	6.47	7.74	5.51	0.555	0.20	7.31	6.12	8.96	5.71	0.413	0.28
EDI-2 Perfectionism	5.48	4.38	6.22	4.59	0.631	0.17	6.15	4.57	5.74	4.53	0.786	0.09
EDI-2 Impulse regulation	4.45	5.01	4.55	4.41	0.951	0.02	2.82	2.32	5.80	5.47	0.043 *	0.71 ^†^
EDI-2 Asceticism	6.14	3.71	5.81	3.21	0.786	0.09	6.33	3.61	5.65	3.25	0.557	0.20
EDI-2 Social insecurity	7.85	5.95	6.38	3.50	0.368	0.30	7.54	5.29	6.54	4.11	0.526	0.21
EDI-2 Total	93.27	41.29	93.95	31.78	0.957	0.02	93.13	37.95	94.09	34.29	0.936	0.03
UPPS-P Lack of premeditation	23.03	6.91	21.30	6.00	0.434	0.27	20.93	6.89	22.81	6.07	0.378	0.29
UPPS-P Lack of perseverance	24.29	5.89	21.76	6.45	0.212	0.41	21.83	6.06	23.51	6.77	0.394	0.26
UPPS-P Sensation seeking	22.89	6.15	26.39	7.84	0.172	0.50 ^†^	24.26	7.37	25.52	7.46	0.617	0.17
UPPS-P Positive urgency	24.66	8.59	27.73	11.06	0.389	0.31	23.42	7.97	28.83	11.07	0.111	0.56 ^†^
UPPS-P Negative urgency	32.70	8.06	34.07	7.27	0.584	0.18	31.22	7.93	35.26	6.91	0.088	0.54 ^†^
UPPS-P Total score	127.59	21.85	131.37	24.78	0.634	0.16	121.66	17.87	136.07	25.65	0.044 *	0.65 ^†^
TCI-R Novelty seeking	94.49	20.70	98.53	15.44	0.505	0.22	87.36	17.07	104.15	14.67	0.002 *	1.05 ^†^
N2 Positive go	−4.00	2.72	−3.50	2.32	0.571	0.20	−4.69	2.71	−2.94	2.01	0.033 *	0.73 ^†^
N2 Positive no-go	−5.08	2.46	−4.03	2.46	0.248	0.43	−5.15	2.47	−3.86	2.25	0.113	0.54 ^†^
N2 Negative go	−4.53	2.53	−4.01	2.28	0.528	0.22	−5.05	2.32	−3.59	2.25	0.063	0.64 ^†^
N2 Negative no-go	−5.12	2.12	−4.47	2.98	0.483	0.25	−5.89	2.58	−3.85	2.40	0.019 *	0.82 ^†^
N2 Neutral go	−4.20	2.38	−3.73	2.06	0.542	0.21	−4.80	2.08	−3.25	2.04	0.031 *	0.75 ^†^
N2 Neutral no-go	−5.48	2.40	−4.43	2.69	0.251	0.41	−5.80	2.82	−4.13	2.19	0.054	0.66 ^†^
Accuracy Negative go	0.99	0.02	0.96	0.06	0.110	0.59 ^†^	0.97	0.04	0.97	0.06	0.904	0.04
Accuracy Negative no-go	0.79	0.12	0.81	0.12	0.663	0.15	0.80	0.10	0.80	0.13	0.956	0.02
Accuracy Positive go	0.99	0.02	0.96	0.06	0.071	0.68 ^†^	0.97	0.06	0.97	0.05	0.656	0.15
Accuracy Positive no-go	0.80	0.11	0.81	0.12	0.827	0.08	0.80	0.09	0.80	0.13	0.981	0.01
Accuracy Neutral go	0.98	0.03	0.95	0.07	0.168	0.52 ^†^	0.96	0.08	0.97	0.04	0.383	0.28
Accuracy Neutral no-go	0.76	0.11	0.80	0.13	0.355	0.33	0.77	0.08	0.80	0.15	0.571	0.20
Stroop interference	8.72	14.57	0.22	11.07	0.036 *	0.66 ^†^	8.20	15.44	0.21	9.46	0.042 *	0.62 ^†^

Note. Good outcome: full remission. Bad outcome: dropout, non-remission or partial remission. SD: standard deviation. * significant comparison. ^†^ Effect size in the ranges mild–moderate to high–large.

**Table 3 nutrients-14-05011-t003:** Predictive models for the risk of poor outcome: logistic regression adjusted for ED subtype.

	B	SE	*p*	OR	95%CI (OR)		HL	NR^2^	AUC
Poor outcome: final treatment									
Stroop interference	–0.076	0.041	0.027	0.927	0.855	0.998	0.941	0.230	0.718
Poor outcome: 2-year follow-up									
TCI-R Novelty seeking	0.078	0.032	0.003	1.082	1.017	1.150	0.090	0.405	0.872
N2 Negative no-go	0.394	0.206	0.027	1.484	1.001	2.220			
Stroop interference	–0.090	0.053	0.040	0.914	0.824	0.999			

Note. SE: standard error. OR: odds ratio. HL: Hosmer–Lemeshow test (*p*-Value). NR^2^: pseudo-R2 coefficient. AUC: area under ROC.

## Data Availability

Data are not available in any repository. Please contact the corresponding authors.

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
