# Peer review of "Impact of Impulsivity and Therapy Response in Eating Disorders from a Neurophysiological, Personality and Cognitive Perspective"

_nutrients, 2022, doi:10.3390/nu14235011_

Round 1

Reviewer 1 Report

First of all, thank you for inviting me to review the article, Impact of impulsivity and therapy response in eating disorders: form a neurophysiological, personality, and cognitive perspective.

Here are some questions and revision recommendations:

Title should be - Impact of impulsivity and therapy response in eating disorders: “From” a neurophysiological, personality, and cognitive perspective, instead of “form”- please recheck entire paper for typing error. Thanks

Abstract should follow journal style; in paragraph form – please remove “background”, “methods”, “results”, “conclusion”

As the author/s noted in line 87 to 89, “research is needed to determine that whether a lack of inhibitory control is related to treatment response…”, is the current study related or answers this gap? Should clarify, which factors within the study are classified as inhibitory controls.

Information regarding the ethical considerations for studies on human subjects should be provided in the materials/method section – eg, IRB, informed consent, voluntary participation,…

Goodness of fit criteria / cutoff scores should be mention in lines 209, also, references should be provided

how about the other background demographics?

Table 2 further clarification needed, which variables are independent, dependent, covariance

Table 2 “P” should be “p”

So what now, discussion should be expanded into to include more practical significance of the findings. 

Author Response

First of all, thank you for inviting me to review the article, Impact of impulsivity and therapy response in eating disorders: form a neurophysiological, personality, and cognitive perspective.

Here are some questions and revision recommendations:

  1. Title should be - Impact of impulsivity and therapy response in eating disorders: “From” a neurophysiological, personality, and cognitive perspective, instead of “form”- please recheck entire paper for typing error. Thanks

RESPONSE: Thank you for the indication. This has been corrected.

  1. Abstract should follow journal style; in paragraph form – please remove “background”, “methods”, “results”, “conclusion”

RESPONSE: Thank you. We adapted the abstract to the journal style.

3. As the author/s noted in line 87 to 89, “research is needed to determine that whether a lack of inhibitory control is related to treatment response…”, is the current study related or answers this gap? Should clarify, which factors within the study are classified as inhibitory controls.

RESPONSE: Thank you for highlighting this point. We assessed different dimensions of inhibitory control and its relation with treatment outcome in EDs to fill the current gap in the literature. We implemented a paragraph to introduce the dimensions of inhibitory control considered and the tasks adopted.  

lines 49-55:

Inhibitory control refers to the ability to inhibit cognitive or motor responses [3–5]. Cognitive inhibition is usually measured by interference control tasks (e.g. Stroop Color-Word Task) in which effortful inhibition at a covert, cognitive level is required to suppress the competing automatic response in favor of the correct response [6]. In contrast, inhibition of motor responses is assessed by go/no-go tasks, which measure the overt, effortful expression of inhibition involving the suppression of activated motor response [6].  

lines 96-101:

“The present study aimed to analyze the impact of several facets of impulsivity in ED’s therapy response. To this scope, multiple components of impulsivity were evaluated in a comprehensive perspective including 1) impulsive personality, measured by the UPPS dimensions and novelty-seeking trait 2) inhibitory control process, measuring cognitive inhibition by a Stroop task, and motor inhibition by an emotional go/no-go task with EEG recording.”

  1. Information regarding the ethical considerations for studies on human subjects should be provided in the materials/method section – eg, IRB, informed consent, voluntary participation,…

RESPONSE: Thank you. We added this information in the method section

lines: 116-117:

“Patients voluntarily participated in the study and their informed consent was collected”

  1. Goodness of fit criteria / cutoff scores should be mention in lines 209, also, references should be provided

RESPONSE: Thank you for pointing this out. We have indicated that adequate goodness-of-fit is achieved for a p>.05 value in the Hosmer-Lemeshow test (line 223). We have also included the following reference:

Hosmer DW, Hosmer T, Le Cessie S, Lemeshow S. A comparison of goodness-of-fit tests for the logistic regression model. Stat Med. 1997;16(9):965-980. doi:10.1002/(sici)1097-0258(19970515)16:9<965::aid-sim509>3.0.co;2-o

  1. how about the other background demographics?

RESPONSE: Thank you for the comment. We reported in table 1 the main sociodemographic variables such as age, education, civil status, employment, and social position. All participants were female with EDs under treatment. No other relevant demographic data were collected in the study.

  1. Table 2 further clarification needed, which variables are independent, dependent, covariance

RESPONSE: Thank you for this feedback. We have now clarified the results of Table 2, corresponding to ANCOVA procedures exploring the relationships between the clinical variables measured at baseline (duration of the ED, EDI-2 scales, UPPS-P scales, TCI-R novelty seeking, and go/no-go tasks) and the treatment outcome at the end of the intervention and at 2-years of the follow-up (good / poor). We have also indicated that these analyses were adjusted-controlled by the ED subtype.

lines (245-249):

“Table 2 shows the results of the ANCOVA exploring the relationships between the clinical variables measured at baseline (duration of the ED, EDI-2 scales, UPPS-P scales, TCI-R novelty seeking, go/no-go task, and Stroop interference) and the treatment outcome (good / poor) measured at two-time points 1) at final treatment, after CTB; 2 ) at 2 years of follow-up. These analyses were adjusted-controlled by the ED subtype)

  1. Table 2 “P” should be “p”

RESPONSE: Thank you. This has been corrected.

  1. So what now, discussion should be expanded into to include more practical significance of the findings.

RESPONSE: Thank you for the suggestion. We implemented the description of practical and clinical implications of these findings.

Lines 334-345:

“Taken together, the present results highlighted the relevance of novelty-seeking and inhibitory control in remission from EDs. A comprehensive assessment of impulsivity, including personality traits and neurocognitive indices of inhibitory control, may be particularly useful to improve treatment effectiveness. For instance, those individuals with the tendency to be more impulsive, excitable, dramatic, and with intolerance to routine might benefit from treatment tailored to reduce their impulsive behaviors. To address difficulties in inhibition, inhibitory control training with general or food-specific stimuli has been tested in individuals with EDs showing promising results [64,65]. Recently, the outcome of a food-specific inhibitory control training has been measured with ERPs indexes [66]. The results from our emotional go/no-go task encourage testing the effectiveness of novel inhibitory training using emotional stimuli to target impulsivity and emotion regulation.”

Reviewer 2 Report

Although this is an interesting paper, there are some points for improvement.

The introduction focuses only on impulsivity in eating disorders, but a broader introduction could be helpful to the reader in order to understand the concept of impulsivity in general, in relation to other psychological parameters, and then more specifically focused on eating disorders. For example, some useful information can be found in recent articles such as:

Rochat, L., Billieux, J., Gagnon, J., & Van der Linden, M. (2018). A multifactorial and integrative approach to impulsivity in neuropsychology: insights from the UPPS model of impulsivity. Journal of Clinical and Experimental Neuropsychology40(1), 45-61.

Stoyanova, S., Ivantchev, N., & Giannouli, V. (2021). Functional, Dysfunctional Impulsivity and Sensation Seeking in Medical Staff. Psychiatria Danubina33(suppl 10), 25-29.

Carr, M. M., Wiedemann, A. A., Macdonald-Gagnon, G., & Potenza, M. N. (2021). Impulsivity and compulsivity in binge eating disorder: A systematic review of behavioral studies. Progress in Neuro-Psychopharmacology and Biological Psychiatry110, 110318.

Stoyanova, S., & Giannouli, V. (2018). Bulgarian students’ impulsivity differentiated by gender, age, and some scientific areas. Psychological Thought11(2).

McDonald, C. E., Rossell, S. L., & Phillipou, A. (2019). The comorbidity of eating disorders in bipolar disorder and associated clinical correlates characterised by emotion dysregulation and impulsivity: a systematic review. Journal of Affective Disorders259, 228-243.

The recruitment of the participants is not clear. In addition to that, the sample size is really small for parametric statistics to perform. A possible solution is to rerun the study and provide a larger sample of participants.

Is there a control group? Please focus on this point.

Why is the Stroop test chosen, instead of other relevant neuropsychological tests?

The tables are difficult to follow, so please use graphs instead.

A more detailed discussion is needed that takes into consideration all parts of the results section.

Author Response

Although this is an interesting paper, there are some points for improvement.

1: The introduction focuses only on impulsivity in eating disorders, but a broader introduction could be helpful to the reader in order to understand the concept of impulsivity in general, in relation to other psychological parameters, and then more specifically focused on eating disorders. For example, some useful information can be found in recent articles such as:

Rochat, L., Billieux, J., Gagnon, J., & Van der Linden, M. (2018). A multifactorial and integrative approach to impulsivity in neuropsychology: insights from the UPPS model of impulsivity. Journal of Clinical and Experimental Neuropsychology40(1), 45-61.

Stoyanova, S., Ivantchev, N., & Giannouli, V. (2021). Functional, Dysfunctional Impulsivity and Sensation Seeking in Medical Staff. Psychiatria Danubina33(suppl 10), 25-29.

Carr, M. M., Wiedemann, A. A., Macdonald-Gagnon, G., & Potenza, M. N. (2021). Impulsivity and compulsivity in binge eating disorder: A systematic review of behavioral studies. Progress in Neuro-Psychopharmacology and Biological Psychiatry110, 110318.

Stoyanova, S., & Giannouli, V. (2018). Bulgarian students’ impulsivity differentiated by gender, age, and some scientific areas. Psychological Thought11(2).

McDonald, C. E., Rossell, S. L., & Phillipou, A. (2019). The comorbidity of eating disorders in bipolar disorder and associated clinical correlates characterised by emotion dysregulation and impulsivity: a systematic review. Journal of Affective Disorders259, 228-243.

RESPONSE: Thank you for your comment. We thank you for the suggested articles, that we have now included in the manuscript. We agree with you that a more general introduction to impulsivity dimensions could help to contextualize. This has been added in the first lines of the introduction (44-55): 

“Impulsivity is recognized as a multidimensional construct reflecting multiple and separable psychological dimensions. An important contribution to this multidimensional conception has been provided by the UPPS model, which describes different personality traits that reflect impulsive behaviors [1].

From a neuropsychological perspective, some of the cognitive functions linked to impulsivity are inhibitory control and decision-making processes [2]. Inhibitory control refers to the ability to inhibit cognitive or motor responses [3–5]. Cognitive inhibition is usually measured by interference control tasks (e.g. Stroop Color-Word Task) in which effortful inhibition at a covert, cognitive level is required to suppress the competing automatic response in favor of the correct response [6]. In contrast, inhibition of motor responses is assessed by go/no-go tasks, which measure the overt, effortful expression of inhibition involving the suppression of activated motor response [6].”  

  1. The recruitment of the participants is not clear. In addition to that, the sample size is really small for parametric statistics to perform. A possible solution is to rerun the study and provide a larger sample of participants.

RESPONSE: Thank you for the comments. We clarified the recruitment procedure in the method section (lines 111-114):

A total of 37 treatment-seeking individuals with ED were consecutively recruited at the ED Unit within the Department of Psychiatry at Bellvitge University Hospital (HUB)—a public health hospital certified as a tertiary care center with a highly specialized unit for the treatment of ED in Barcelona (Spain).”

In response to the reviewer, we have now added in the limitations subsection that the small sample size impacts the ability to avoid Type II errors, the power capacity to detect the existence of true relationships, and the accuracy of the results obtained in the multivariate analyses. In this sense, the empirical evidence of this work should be interpreted with caution, pending future studies with larger samples to confirm or refute them.

Lines 350-354:

“The small sample size also impacts the ability to avoid Type II errors, the power capacity to detect the existence of a true relationship, and the accuracy of the results obtained in the multivariate analyses. In this sense, the empirical evidence of this work should be interpreted with caution, pending future studies with larger samples to confirm or to refute them.”

  1. Is there a control group? Please focus on this point.

RESPONSE: Thank you for this comment. Unfortunately, it was not possible to include a control group of patients not receiving the CBT treatment or receiving others types of treatment. This has been added as a limitation (lines 362-363):

“Finally, the absence of a control group of patients undergoing treatment other than CBT or untreated individuals limits the interpretation and generalization of these results.”

  1. Why is the Stroop test chosen, instead of other relevant neuropsychological tests?

RESPONSE: Thank you for raising this question. First, we considered that a more comprehensive way to study subcomponents of inhibitory control would be to evaluate both motor and cognitive inhibition. For this reason, in addition to the go/no-go for motor inhibition, we decided to use the Stroop task, which is a reliable and solid instrument to assess cognitive inhibitory processes.

A second reason to use the Stroop task derives from previous literature. For instance, poor interference control in the Stroop was associated with longer EDs duration (Miranda-Olivos et al. 2020), which in turn is a factor contributing to poor remission. However, there is a lack of studies analyzing the direct relationship between cognitive inhibition and remission from EDs symptomatology, therefore we tried to fill this gap.

This concept and the aforementioned reference have been added to the discussion (lines 314-319):

“reduced cognitive inhibition has been associated with longer EDs duration in patients with AN [59], which in turn is a factor contributing to poor remission from EDs. The present results in a mixed sample of EDs suggest that cognitive inhibition is a relevant factor associated with both short-term and long-term remission. Future studies in larger samples would help to detect and elucidate differences across EDs subtypes.

  1. The tables are difficult to follow, so please use graphs instead.

RESPONSE: Thank you for this recommendation. We have now included a new figure with two radar charts including the z-standardized means for the groups defined based on the treatment outcomes at the end of the treatment and at 2-years of the follow up (new Figure 2).

  1. A more detailed discussion is needed that takes into consideration all parts of the results section.

RESPONSE: Thank you for the suggestion. We restructured part of the discussion to give a more detailed interpretation of the current results and their potential clinical implications.

Results are discussed more in deep, such as in lines 320-325:

“Regarding motor inhibition, the lower amplitude of the N2 was associated with poor ED remission at follow-up. By contrast, behavioral measures of response accuracy in the go/no-go did not predict EDs remission, in line with some previous findings [25]. It can be argued that ERPs indexed such as the N2, may be particularly sensitive in detecting alterations in inhibitory processes, as recently showed in patients with BN undergoing an odd-ball task [60]”.

In response to your comment, and as suggested also by reviewer 1, we improved the discussion of the overall findings of this study, adding also some practical implications (lines 334-345):

Taken together, the present results highlighted the relevance of novelty-seeking and inhibitory control in remission from EDs.A comprehensive assessment of impulsivity, including personality traits and neurocognitive indices of inhibitory control, may be particularly useful to improve treatment effectiveness. For instance, those individuals with the tendency to be more impulsive, excitable, dramatic, and with intolerant of routine might benefit from treatment tailored to reduce their impulsive behaviors. To address difficulties in inhibition, inhibitory control training with general or food-specific stimuli has been tested in individuals with EDs showing promising results [64,65]. Recently, the outcome of a food-specific inhibitory control training has been measured with ERPs indexes [66]. The results from our emotional go/no-go task encourage testing the effectiveness of novel inhibitory training using emotional stimuli to target impulsivity and emotion regulation.”
